# Effects of Losartan, Atorvastatin, and Aspirin on Blood Pressure and Gut Microbiota in Spontaneously Hypertensive Rats

**DOI:** 10.3390/molecules28020612

**Published:** 2023-01-06

**Authors:** Shuai Dong, Qi Liu, Xue Zhou, Yubo Zhao, Kang Yang, Linsen Li, Dan Zhu

**Affiliations:** 1School of Pharmacy, Minzu University of China, Beijing 100081, China; 2Chengdu Institute of Biology, Chinese Academy of Sciences, Chengdu 610041, China

**Keywords:** gut microbiota, cardiovascular disease, losartan, atorvastatin, aspirin

## Abstract

Many studies have shown that alterations in the gut microbiota are associated with hypertension. Our study aimed to observe the characteristics of the gut microbiota in hypertension and to further explore whether drug molecules can play a therapeutic role in hypertension by interfering with the gut microbiota. We evaluated the differences in the composition of the gut microbiota in spontaneously hypertensive rats (SHR) and Wistar Kyoto rats (WKY). Meanwhile, three first-line cardiovascular disease (CVD) drugs, losartan, atorvastatin, and aspirin, were used to treat the SHR in order to observe their effects on the gut microbiota in SHR. The 16S rDNA results showed that the diversity and richness of the gut microbiota in SHR were significantly reduced compared with that of the WKY, the Firmicutes/Bacteroidetes ratio was increased, the abundances of *Bifidobacterium* and short chain fatty acids (SCFAs)-producing bacteria decreased, and the abundance of lactate-producing bacteria increased. In addition to lowering the blood pressure, losartan increased the abundances of *Alistipes*, *Bacteroides,* and *Butyricimonas* in SHR, reduced the abundances of *Ruminococcaceae*, *Streptococcus,* and *Turicibacter*, reduced the Firmicutes/Bacteroidetes ratio, and rebalanced the gut microbiota. Losartan also increased the abundances of *Bifidobacterium* and SCFAs-producing bacteria and reduced the abundance of lactate-producing bacteria. However, atorvastatin and aspirin had no significant effect on the gut microbiota in SHR. The above results showed that losartan could change the characteristics of the gut microbiota in hypertension and rebalance the gut microbiota, which may be related to lowering the blood pressure. Atorvastatin and aspirin have no significant influence on the gut microbiota in SHR.

## 1. Introduction

In recent years, the potential importance of the gut microbiota to human health has received extensive attention. Gut microbiota, which is known as the second human genome, has become one of the most popular research fields [1]. The human gut microbiota consists of trillions of microorganisms, which are dominated by four phyla: Firmicutes, Bacteroidetes, Actinobacteria, and Proteobacteria [2]. Firmicutes and Bacteroidetes are the dominant bacteria in humans, and there are increasing evidence to suggest that changes in the ratio of Firmicutes (F) to Bacteroidetes (B) in the microbial community, which is known as the F/B ratio, can be used as a biomarker for a pathological condition [3,4]. The gut microbiota and the human body are interdependent, coordinate, and symbiotic, and they synergistically regulate the physiological activities of the organism [5]. The survival of the gut microbiota depends on the rich nutrition and relatively stable environment provided by the human body. The gut microbiota can participate in the absorption and metabolism of nutrients in the body, maintain the integrity of the intestinal mucosal barrier, and the balance of tissue homeostasis, and also maintain immune homeostasis and prevent the invasion of pathogenic bacteria, which is an important guarantee for human health [6,7]. Having a stable balance in the gut microbiota composition is the key to maintaining intestinal immunity and whole-body homeostasis [8,9].

Cardiovascular disease (CVD) is the main public health care project of the World Health Organization (WHO) [10]. With the rapid development of molecular biology and bioinformatics, an increasing number of findings have shown that alterations to the gut microbiota are associated with CVDs such as atherosclerosis, coronary heart disease, hypertension, and heart failure [11,12,13]. When the composition and structure of the gut microbiota in the human body changes, it causes intestinal malnutrition, inflammation, and an abnormal metabolism, thereby promoting the development of hypertension and atherosclerosis [14,15]. Hypertension is a major risk factor for CVD, but the characteristics of the gut microbiota in hypertension are currently unclear, which is necessary for further exploration of the relationship between the gut microbiota and hypertension.

Some studies have shown that some CVD drugs changed the gut microbiota. Recent studies have shown that irbesartan reduced the F/B ratio in the gut microbiota in the high-altitude pulmonary hypertension rats and improved the disorder of the gut microbiota [16]. Two studies have shown that atorvastatin and aspirin also changed the composition of the gut microbiota, which may be a new target for them to play a role in treatments [17]. However, the characteristics of the gut microbiota in the disease still need to be further clarified. Meanwhile, more evidence is still needed to demonstrate whether drugs can play a therapeutic role in CVD by improving the dysbiosis of the gut microbiota. Therefore, our study took spontaneously hypertensive rats (SHR) as the observation object to explore the characteristics of the gut microbiota in hypertension and investigated the effect of three cardiovascular therapeutic drugs, losartan, atorvastatin, and aspirin, on the gut microbiota in SHR. Our study hopes to provide evidential support for drugs to play a therapeutic role by interfering with the gut microbiota.

## 2. Results

### 2.1. Blood Pressure

Over time, the systolic blood pressure (SBP) of Wistar Kyoto rats (WKY) was maintained at about 140 mmHg, and the diastolic blood pressure (DBP) was maintained at about 100 mmHg. The SBP of SHR was maintained at about 210 mmHg, and the DBP was maintained at about 150 mmHg. The SBP and DBP of SHR were significantly higher than those of WKY (*p* < 0.01). After four weeks of a losartan treatment, the SBP and DBP of SHR were significantly decreased (*p* < 0.01). After the atorvastatin and aspirin treatments, the SBP and DBP of SHR did not change significantly (Figure 1).

### 2.2. Quality Control of Sequencing Data

As shown in Figure 2, 2,281,976 effective sequences were detected in all of the samples, and the average effective length of the sequences was about 418 bp. There were no significant differences in the quality control results of the sequencing data of the gut microbiota between the five groups of samples (*p* > 0.05). The sequencing coverages of the gut microbiota in all of the rats were above 0.999, which indicated that the sequences had a high probability of being sequenced. In conclusion, the sequencing results could be used for a subsequent bioinformatics analysis.

### 2.3. Operational Taxonomic Units (OTUs) Analysis

All of the quality sequences were divided into 606 OTUs based on a 97% similarity level. A Venn diagram was drawn to observe the similarity and overlapping of the five groups of OTUs (Figure 3). Our results showed that the number of OUTs for the WKY, SHR, SHR-Losartan (Lo), SHR-Atorvastatin (At), and SHR-Aspirin (As) groups were 544, 603, 597, 553, and 551, respectively. Four hundred and seventy-seven OTUs were shared by the five groups, indicating that the similarity of OTUs among the five groups was high.

### 2.4. Alpha and Beta Diversity of Gut Microbiota

After obtaining the OTUs abundance array, the Alpha and Beta diversity analyses were performed on the gut microbiota in all of the rats. The composition of the gut microbiota in the rats was assessed by calculating four main ecological parameters, including an abundance coverage-based estimator (ACE), Chao, Simpson, and Shannon indices. ACE and Chao are positive relationship to the richness of the gut microbiota. Simpson and Shannon have a positive relationship with the diversity of the gut microbiota. Our results showed that the ACE, Chao, Simpson, and Shannon indices in the SHR was significantly lower than those in the WKY (*p* < 0.01). This indicated that the diversity and richness of the gut microbiota in the SHR were significantly reduced. After the losartan treatment, the diversity and richness of the gut microbiota in the SHR were significantly increased (*p* < 0.01). After the atorvastatin and aspirin treatments, the diversity and richness of the gut microbiota in the SHR did not change significantly (Figure 4A–D).

In this study, a principal co-ordinates analysis (PCoA) and a nonmetric multidimensional scaling (NMDS) analysis were performed based on the Bray–Curtis distance. Our results show that the samples from the WKY and SHR were farther apart and completely separated, indicating that the compositions of the gut microbiota were significantly different between the two groups of samples. The distances between the samples in the SHR, At, and As groups were small, indicating that the three groups of samples were very similar in composition. Excitingly, the samples in the Lo and SHR groups were significantly dispersed, and the distances between the samples in the Lo and WKY groups were smaller. These indicated that the gut microbiota in the SHR was significantly changed after the losartan treatment, and it was very similar to the gut microbiota in the WKY (Figure 5).

### 2.5. Structural Analysis of Gut Microbiota

Firmicutes and Bacteroidetes accounted for more than 90% of the gut microbiota in the rats. An increased F/B ratio is now widely regarded as a hallmark of dysbiosis. To demonstrate whether there is a similar ratio in the gut microbiota in the SHR, we analyzed the abundance of each phylum. Our results showed that the fecal samples from the rats were dominated by Firmicutes and Bacteroidetes, with smaller proportions of Actinobacteria, Proteobacteria, and Tenericutes (Figure 6A). Our results showed that the abundances of Actinobacteria and Bacteroidetes in the SHR were significantly lower than those in the WKY (*p* < 0.01), while the abundance of Firmicutes in SHR was significantly increased (Table 1 and Figure 6A). There was no significant difference in the abundances of Proteobacteria and Tenericutes in the WKY and SHR, which indicated that these two phyla are not characteristics of the gut microbiota in hypertension (Table 1 and Figure 6A). Excitingly, after the losartan treatment, the abundances of Actinobacteria and Bacteroidetes in the SHR were significantly increased (*p* < 0.01), and the abundance of Firmicutes was significantly decreased (*p* < 0.01). Compared with the WKY, the F/B ratio in the SHR group increased 1.3 times (*p* < 0.01) (Figure 6B). Compared with the SHR, the F/B ratio in the Lo group was significantly decreased. The atorvastatin and aspirin treatment did not change composition of the gut microbiota. These results suggested that losartan could improve the disorder of the gut microbiota in the SHR and rebalance the gut microbiota.

Considering the imbalanced F/B ratio, we explored what bacterial genera led to the changes in the composition of the gut microbiota in the SHR. The linear discriminant analysis effect size (LEfSe) performed at the genus level showed that the abundances of *Alistipes*, *Bacteroides*, and *Butyricimonas* belonging to Bacteroidetes were significantly enriched in the WKY, while these three genera were significantly decreased in the SHR, At, and As groups (Figure 7A). Interestingly, after the losartan treatment, the abundances of these three genera in the SHR were significantly increased (Figure 7B). The abundances of *Dubosiella*, *Ruminococcaceae*, *Ruminococcus*, *Streptococcus*, and *Turicibacter,* which belong to Firmicutes, were significantly increased in the SHR, while the abundances of these five genera was significantly decreased in the WKY (Figure 7A). After the losartan treatment, the abundances of *Ruminococcaceae*, *Streptococcus*, and *Turicibacter* were significantly decreased (Figure 7B). These changes are the main cause for the increase in the F/B in the SHR. What is exciting is that losartan changed the composition of the gut microbiota, reduced the F/B ratio in the SHR, and rebalanced the gut microbiota.

As shown in Figure 8A,B, the abundances of *Alistipes* and *Bacteroides* in the SHR were significantly lower than those in the WKY (*p* < 0.01). After the losartan treatment, the abundances of these two genera in the SHR significantly increased (*p* < 0.01). However, compared with the SHR, the abundances of *Alistipes* and *Bacteroides* in the At and As groups did not change significantly. *Bifidobacterium*, which belongs to the phylum of Actinobacteria, is commonly considered to be a beneficial genus of bacteria, and it plays a key role in the immune system [18]. It is reported that the abundance of *Bifidobacterium* was reduced under multiple disease conditions, which is also a feature of dysbiosis [19,20]. Our results showed that the abundance of *Bifidobacterium* in the SHR was significantly lower than that in the WKY (*p* < 0.01). This greatly reduced the proportion of Actinobacteria in the SHR, thereby reducing the diversity of the gut microbiota. After the losartan treatment, the abundance of *Bifidobacterium* in the SHR significantly increased (*p* < 0.01). However, the abundance of *Bifidobacterium* in the At and As groups did not change significantly (Figure 8C). The abundance of *Butyricimonas* in the SHR was significantly lower than that in the WKY (*p* < 0.01) (Figure 8D). After the losartan treatment, the abundance of *Butyricimonas* in the SHR significantly increased (*p* < 0.01). The abundances of *Dubosiella* and *Ruminococcus* in SHR were significantly higher than those in the WKY (*p* < 0.01) (Figure 8E,G). However, compared with the SHR, the abundances of *Dubosiella* and *Ruminococcus* in the Lo, At, and As groups did not change significantly. The abundances of *Ruminococcaceae*, *Streptococcus*, and *Turicibacter* in the SHR were significantly higher than those in the WKY (*p* < 0.01) (Figure 8F,H,I). After the losartan treatment, the abundances of these three genera in the SHR significantly decreased (*p* < 0.01). It can be seen more intuitively from Figure 8J that losartan increased the abundance of *Alistipes*, *Bacteroides*, *Bifidobacterium*, and *Butyricimonas* in the SHR, reduced the abundance of *Ruminococcaceae*, *Streptococcus*, and *Turicibacter*, and rebalanced the gut microbiota.

### 2.6. The Abundance of Acetate-, Butyrate-, and Lactate-Producing Bacteria

LEfSe also showed a higher abundance of acetate- and butyrate-producing bacteria in the WKY, including *Bacteroides* and *Butyricimonas*. In contrast, a high abundance of lactate-producing bacteria was found in the SHR, including *Streptococcus* and *Turicibacter*. To further clarify the changes of short chain fatty acids (SCFAs)- and lactate-producing bacteria in rats of each group, we re-grouped all of the bacterial 16S rDNA reads according to their major metabolic end-products, as described in the methods. Our results showed that the abundances of acetate- (Figure 9A) and butyrate-producing bacteria (Figure 9B) in the SHR were significantly lower than those in the WKY (*p* < 0.01), while the abundance of lactate-producing bacteria (Figure 9C) in the SHR was significantly increased (*p* < 0.01). Obviously, after the losartan treatment, the abundances of acetate- and butyrate-producing bacteria in the SHR significantly increased (*p* < 0.01), and the abundance of lactate-producing bacteria significantly decreased (*p* < 0.01).

## 3. Discussion

CVD is one of the leading causes of death worldwide. With the rapid development of molecular biology and bioinformatics, more and more studies have shown that alterations to the gut microbiota are associated with CVDs such as atherosclerosis, coronary heart disease, and hypertension. Yang found that the diversity of the gut microbiota decreased, and the F/B ratio increased in both the SHR and hypertensive patients [21]. In another experiment, Yang further found that Minocycline reduced the blood pressure in angiotensin II-induced hypertensive rats, increased the diversity of gut microbiota, and reduced the F/B ratio [21]. These suggest that hypertension is associated with the imbalance of the gut microbiota and that regulating gut microbiota may be a valuable goal of hypertension treatments in the future. Adnan transplanted the gut microbiota in spontaneously hypertensive stroke-prone rats (SHRSP) and WKY with normal blood pressure to each other, and the blood pressure of rats changed in both of the groups [22]. This showed that gut microbiota plays a potential role in the formation of hypertension. However, the characteristics of the gut microbiota in hypertension still need to be further clarified. Meanwhile, more evidence is still needed to demonstrate whether the drugs can play a therapeutic role in hypertension by improving the dysbiosis of the gut microbiota. Therefore, we chose SHR as the observation object to explore the characteristics of the gut microbiota in hypertension, and we investigated the effect of three cardiovascular therapeutic drugs, losartan, atorvastatin, and aspirin, on the gut microbiota in SHR.

The richness and diversity of the gut microbiota are important indicators to reflect the composition of the gut microbiota [23], which can be used to present the characteristics of the gut microbiota under the condition of hypertension. In this study, ACE, Chao, Simpson, and Shannon indices were used to evaluate the richness and diversity of the gut microbiota [24]. Our results show that the diversity and richness of the gut microbiota in the SHR was significantly lower than those in the WKY, which is consistent with previous reports [25]. The results of the PCoA analysis and the NMDS analysis showed that the samples from the WKY and SHR were farther apart and completely dispersed, indicating that the compositions of the gut microbiota were significantly different between the two groups of samples.

A large number of studies have shown that the F/B ratio is an important indicator to reflect the disturbance of the gut microbiota [26]. Our results show that the dominant phyla in the gut microbiota in rats were Firmicutes and Bacteroidetes, which are similar to those of humans. Compared with the WKY, the F/B ratio in the SHR group increased 1.3 times, which indicates that the gut microbiota in SHR is disturbed. LEfSe showed that the abundances of *Alistipes*, *Bacteroides*, and *Butyricimonas* belonging to Bacteroidetes were significantly increased in the WKY, while these three genera were significantly decreased in the SHR. The abundances of *Dubosiella*, *Ruminococcaceae*, *Ruminococcus*, *Streptococcus*, and *Turicibacter,* which belong to Firmicutes, were significantly increased in the SHR, while the abundances of these five genera were significantly decreased in the WKY. These changes are the main cause of the F/B increase in the SHR.

*Alistipes* is associated with many traditional CVDs, such as hypertension, atrial fibrillation (AF), congestive heart failure (CHF), and atherosclerosis [27]. However, it is still unclear whether it is beneficial to CVD. Li found that the diversity of the gut microbiota in hypertensive patients was significantly lower than that in healthy controls, and the abundance of *Bacteroides* related to health was significantly decreased [28]. Kim found that the abundance of butyrate-producing bacteria in hypertensive patients was significantly reduced, such as *Butyricimonas* and *anaerobic Corynebacterium* [29]. It has been found that the abundances of *Ruminococcaceae*, *Streptococcus*, and *Turicibacter* are positively correlated with blood pressure [21,30,31]. Our research results show that the abundances of *Bacteroides* and *Butyricimonas* were significantly decreased in SHR, and they were negatively correlated with blood pressure. We also found that the abundances of *Ruminococcaceae*, *Streptococcus*, and *Turicibacter* were significantly increased in SHR, which were positively correlated with blood pressure. These findings are consistent with previous reports. In addition, we found that the abundance of *Alistipes* was decreased in SHR, which was negatively correlated with blood pressure.

*Bifidobacterium* is commonly considered to be a beneficial genus of bacteria, and it plays a key role in the immune system [32,33,34]. It is reported that the abundance of *Bifidobacterium* is reduced in many diseases, which is also a feature of dysbiosis [35]. Our study found that the abundance of *Bifidobacterium* was significantly decreased in the SHR, further indicating that the gut microbiota in the SHR was disordered.

SCFAs are the main metabolites produced in the colon by the bacterial fermentation of dietary fibers and resistant starch, mainly including acetate, propionate, and butyrate [36,37]. SCFAs play an important role in the development of hypertension, and more and more studies have found that SCFAs can bind to G protein-coupled receptor 41 (GPR41) to reduce the host’s blood pressure [38,39,40]. Several studies have shown that hypertension is associated with SCFAs-producing and lactate-producing bacteria. Yan found that the amounts of SCFAs-producing bacteria such as *Roseburia* and *Faecalibacterium prausnitzii* in the feces of hypertensive patients were lower than those of normal people, and the levels of SCFAs in the feces of hypertensive people were significantly higher than those of normal people [41]. Adnan reported that the abundance of SCFAs-producing bacteria such as *Clostrideae* and *Coprobacillus* was strongly negatively correlated with the increasing systolic blood pressure of rats, while the abundance of lactate-producing bacteria such as *Lactobacillus* was strongly positively correlated with the increasing systolic blood pressure of rats [22]. Our results showed that the abundance of acetate- and butyrate-producing bacteria in SHR was significantly lower than that in WKY, while the abundance of lactate-producing bacteria in SHR was significantly increased, which is consistent with the previous report.

After the losartan treatment, the blood pressure of the SHR was significantly reduced, and the diversity and richness of gut microbiota was significantly increased, and the composition of gut microbiota was very similar to that in the WKY. Losartan increased the abundances of *Alistipes*, *Bacteroides* and *Butyricimonas* in the SHR, reduced the abundances of *Ruminococcaceae*, *Streptococcus* and *Turicibacter*, reduced the F/B ratio, and rebalanced the gut microbiota. Losartan also increased the abundances of *Bifidobacterium* and SCFAs-producing bacteria, and it reduced the abundance of lactate-producing bacteria. The above results suggested that losartan could alter the characteristics of the gut microbiota in hypertension, but more evidence is needed to prove whether it is related to lowering blood pressure. Our study also suggests that the gut microbiota may become a new potential drug target for the treatment of hypertension in the future.

A recent study showed that atorvastatin could increase the abundances of *Bacteroides*, *Butyricimonas*, and *Mucispirillum* in an aged mouse model of high-fat-diet-induced obesity, and it demonstrated that this may contribute to hyperglycemia improvement [42]. Another preliminary study showed that aspirin could change the composition and bacterial taxa of the gut microbiota associated with a colorectal cancer risk [43]. However, our results indicated that atorvastatin and aspirin had no significant effect on the gut microbiota of SHR. Although these two drugs were able to regulate the composition of the gut microbiota in other diseases, their effect on the gut microbiota in hypertension was less pronounced.

## 4. Materials and Methods

### 4.1. Drugs and Reagents

Losartan was purchased from Merck Sharp & Dohme Co., Ltd (Cramlington, UK). Atorvastatin was purchased from Pfizer Pharmaceutical Co., Ltd (New York, NY, USA). Aspirin was purchased from Bayer Healthcare (Beijing, China) Co., Ltd. (Beijing, China). E.Z.N.A.^®^ soil DNA Kit and Phusion^®^ High-Fidelity PCR Master Mix with GC Buffer were purchased from New England Biolabs (Beijing) Co., Ltd. (Beijing, China). Qiagen Gel Extraction Kit was purchased from Shanghai Yubo Biotechnology Co., Ltd. (Shanghai, China). Ion Plus Fragment Library Kit 48 rxns was purchased from Thermo Fisher Scientific (CHINA) Co., Ltd. (Shanghai, China).

### 4.2. Animal Experiments and Design

Six-week-old male SHR (130–150 g) and five-week-old male Wistar Kyoto rats (WKY) (130–150 g) were purchased from Charles River, Inc (Beijing, China, SCXK (Jing) 2016-0006). All of the animals were housed under controlled light (12 h/12 h light–dark cycle), a humidity of 55% ± 5%, and a temperature of 22.1 °C ± 0.5 °C, with free access to water and standard chow. There was one week of adaptive feeding before the formal experiment began. All of the experimental procedures were approved by the Biological and Medical Ethics Committee, Minzu University of China (approval No. ECMU201807).

Thirty-two SHRs were divided into four groups: SHR (vehicle), SHR-Losartan (Lo, 20 mg/kg/day), SHR-Atorvastatin (At, 10 mg/kg/day), and SHR-Aspirin (As, 10 mg/kg/day) ones. All of these drugs and double-distilled water as the vehicle were administrated daily to the SHRs via an oral gavage for 12 weeks. The WKY were treated with double-distilled water as they were a normal control group.

### 4.3. Measurement of Blood Pressure

The SBP and DBP were measured using a noninvasive computerized tail-cuff system (Softron BP-2010, Softron Co., Ltd., Tokyo, Japan). After a pre-warming period under constant temperature in a quiet environment, all of the rats were placed in a restraining chamber. The blood pulse pressure-sensing device was placed at the base of rat tails. Their blood pressures were measured 5–7 times for each rat. The ones that met the compliance were used for determining the average value.

### 4.4. Collection of Fecal Samples

Twelve h after the last administration, fresh feces of all of the rats were collected and placed in sterile EP tubes. The fresh samples were frozen in liquid nitrogen and stored at −80 °C until the analysis.

### 4.5. 16S rDNA High-Throughput Sequencing

Microbial DNA was extracted from the fecal samples using the E.Z.N.A.^®^ soil DNA Kit according to manufacturer’s protocols. The quality of the extracted DNA was checked using 1% agarose gel electrophoresis, while the concentration and purity of DNA were measured using a NanoDrop2000 Spectrophotometer. Then, an appropriate sample of DNA was applied to the centrifuge tube and diluted with sterile water to 1 ng/μL. Using the diluted DNA as a template, primers of the v3–v4 region of 16S rDNA with barcode (338F-5′-ACTCCTACGGGAGGCAGCAG-3′ and 806R-5′-GGACTACHVG GGTWTCTAAT-3′) were used, and a Phusion^®^ High-Fidelity PCR Master Mix with a GC Buffer and high-fidelity thermostable DNA polymerase were used for polymerase chain reaction (PCR) to ensure amplification efficiency and accuracy. For the PCR conditions, refer to previous studies [44]. The PCR products were detected by electrophoresis with 2% agarose gel, and then, they were purified by a Qiagen Gel Extraction Kit. Ion Plus Fragment Library Kit 48 rxns was used to generate the sequencing library. The constructed library passed the Qubit quantitative and library test, and the library was sequenced on Illumina HiSeq 2500 platform (Exxon (Beijing, China) Technology Co., Ltd.).

### 4.6. Bioinformatics Analysis

Paired-end reads of each sample were merged by FLASH (Version 1.2.7, Adobe) to obtain raw tags. High-quality clean tags were generated from the strict filtration of raw tags by QIIME (Version 1.9.1, Rob Knight). Final effective tags were produced after the removal of chimera sequences in the tags, which were detected by the gold database (Version 1.9.1, DOE-JGI) and UCHIME Algorithm. Sequences with 97% homology were clustered into identical OTUs by USEARCH (Version 10.0, Edgar Robert). Then, the representative sequence of each OTU was selected for the species annotation. Raw data see Appendix A. ACE, Chao, Simpson, and Shannon indices were calculated on QIIME [45]. The ade 4 package and ggplot 2 package of R software were used for PCoA, and the vegan software package was used for NMDS analysis [46]. LEfSe was performed online (https://bioincloud.tech/ accessed on 1 June 2022), and the LDA score screening value was set to 3.5. The bacteria were classified according to the lactate and SCFAs end-product as previously described [47,48]. Genera that were defined as producing propionate constituted only a minor portion of the population and were therefore excluded from this analysis. The main experimental process of this study is shown in Figure 10 below.

### 4.7. Statistical Analysis

The statistical analyses were conducted using SPSS Statistics 25 software(International Business Machines Corporation). Differences between the groups were identified using a one-way analysis of variance (ANOVA), which was followed by an LSD post hoc test. Comparisons between the 5 groups in this study are presented as mean ± standard deviation (SD). A value of *p* < 0.05 was considered to be significant.

## 5. Conclusions

In conclusion, we used a 16S rDNA high-throughput sequencing technique to provide the characteristics of the gut microbiota in spontaneously hypertensive rats, mainly determining that: (1) the diversity and abundance of the bacteria decreased significantly, the abundances of *Alistipes*, *Bacteroides*, and *Butyricimonas* increased, the abundances of *Dubosiella*, *Ruminococcaceae*, *Ruminococcus*, *Streptococcus*, and *Turicibacter* decreased, and the F/B ratio increased; (2) the abundance of *Bifidobacterium* decreased, and the abundance SCFAs-producing bacteria decreased, and the abundance lactate-producing bacteria increased. Losartan is an angiotensin II type 1 receptor (AT1R) antagonist, which can change the characteristics of the gut microbiota in hypertension and rebalance the gut microbiota. However, more evidence is needed to prove the relevance to the role of losartan in lowering blood pressure. On the other hand, 12 weeks of administration of atorvastatin or aspirin had no significant influence on the gut microbiota in the SHR.

## Figures and Tables

**Figure 1 molecules-28-00612-f001:**
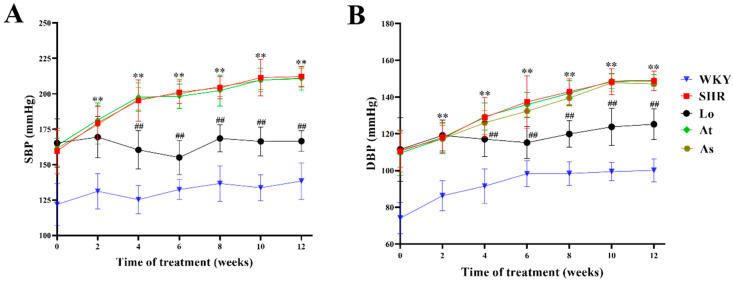
Blood pressure changes in WKY, SHR, SHR-Losartan (Lo), SHR-Atorvastatin (At), and SHR-Aspirin (As) groups. (**A**) Systolic blood pressure (SBP). (**B**) Diastolic blood pressure (DBP). The data are presented as the mean ± standard deviation (SD), n = 6. ** *p* < 0.01 vs. WKY; **^##^** *p* < 0.01 vs. untreated SHR.

**Figure 2 molecules-28-00612-f002:**
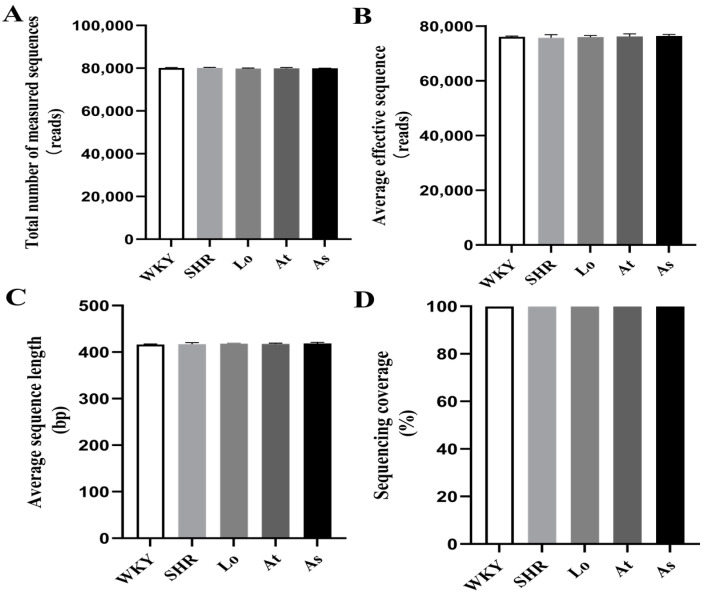
Quality control of sequencing data. (**A**) Total number of measured sequences. (**B**) Average effective sequence. (**C**) Average sequence length. (**D**) Sequencing coverage. The data are presented as the mean ± SD, n = 6.

**Figure 3 molecules-28-00612-f003:**
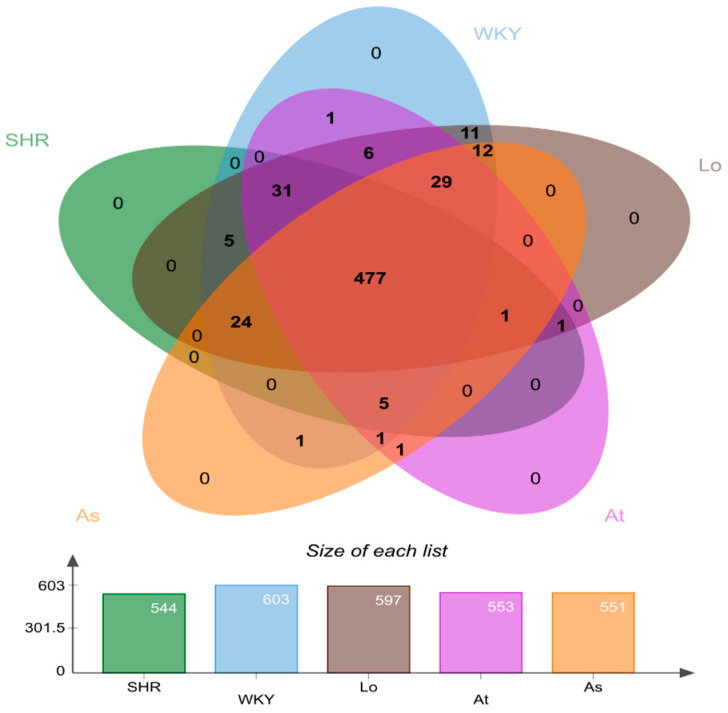
Venn plot of operational taxonomic units (OTUs) distribution of 5 groups of samples. Different groups are represented by different colors, and the overlapping number of different color graphs is the number of common features between groups.

**Figure 4 molecules-28-00612-f004:**
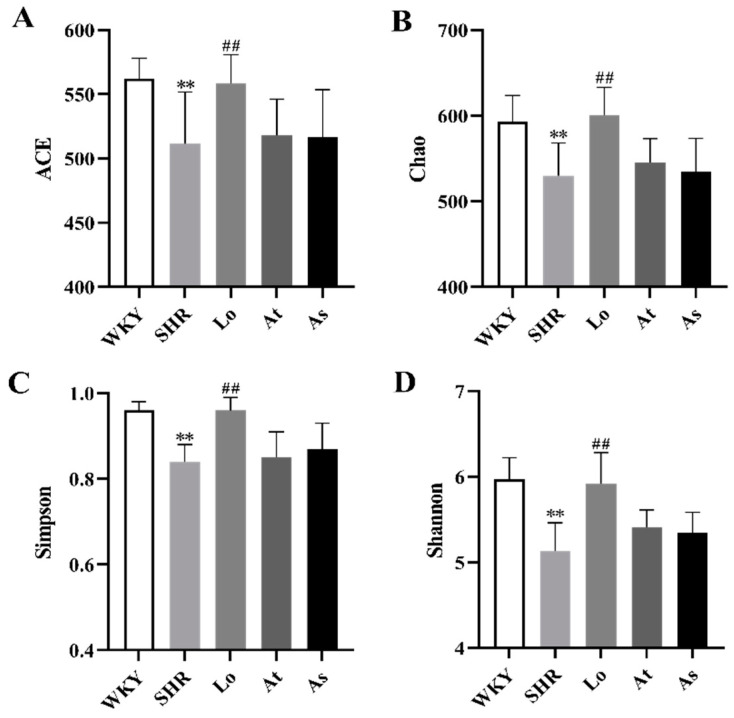
Alpha diversity analysis. (**A**) Abundance coverage-based estimator (ACE). (**B**) Chao index. (**C**) Simpson index. (**D**) Shannon index. The data are presented as the mean ± SD, n = 6. ** *p* < 0.01 vs. WKY; **^##^** *p* < 0.01 vs. untreated SHR.

**Figure 5 molecules-28-00612-f005:**
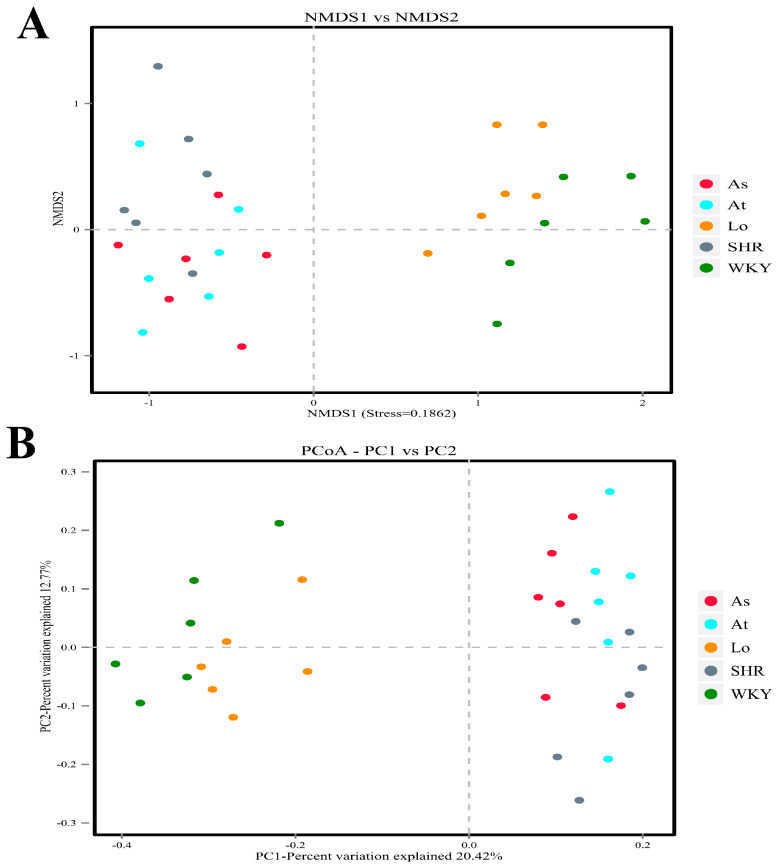
Beta diversity analysis. (**A**) Nonmetric multidimensional scaling (NMDS) is used to compare the differences between sample groups. Each point in the diagram represents a sample. The closer the distance is between the two points, the smaller the difference in the composition of the gut microbiota is. (**B**) Principal co-ordinates analysis (PCoA) was used to investigate the community similarity of gut microbiota. The closer the distance is between the two points, the higher the similarity in the composition of the gut microbiota is.

**Figure 6 molecules-28-00612-f006:**
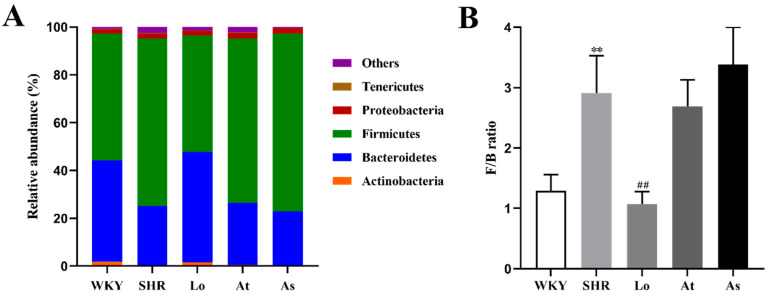
The abundances of the five most abundant phyla in the gut microbiota in rats and F/B ratio. (**A**) The abundances of the five most abundant phyla in the gut microbiota in rats. (**B**) The F/B ratio. The data are presented as the mean ± SD, n = 6. ** *p* < 0.01 vs. WKY; **^##^** *p* < 0.01 vs. untreated SHR.

**Figure 7 molecules-28-00612-f007:**
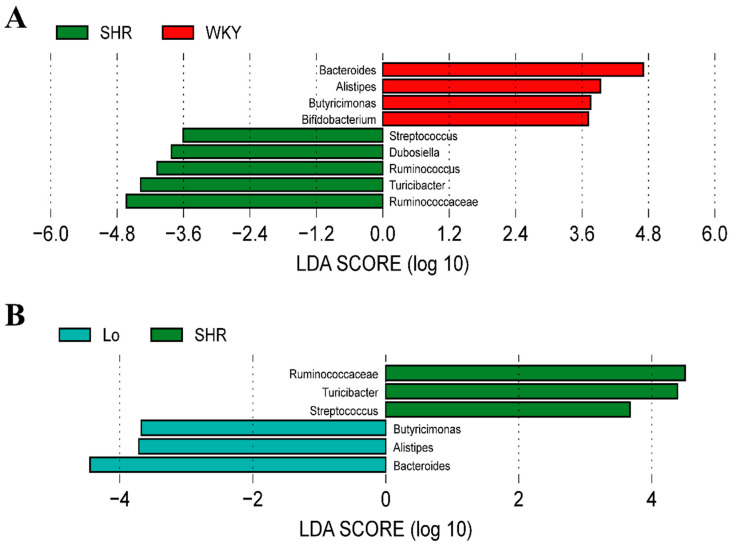
The histogram of the linear discriminant analysis (LDA) value distribution at genus level of gut microbiota, which was used to find biomarkers with statistical differences. (**A**) Bacterial genera with significant difference in abundance between WKY and SHR groups. (**B**) Bacterial genera with significant difference in abundance between SHR-losartan and SHR groups.

**Figure 8 molecules-28-00612-f008:**
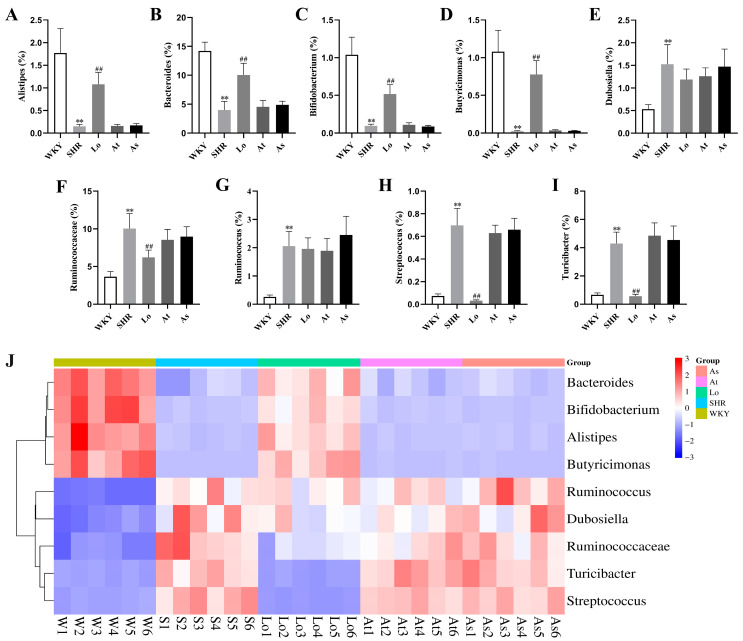
Bacterial genera with significant differences in terms of abundance. (**A**) The abundance of *Alistipes* in rats. (**B**) The abundance of *Bacteroides* in rats. (**C**) The abundance of *Bifidobacterium* in rats. (**D**) The abundance of *Butyricimonas* in rats. (**E**) The abundance of *Dubosiella* in rats. (**F**) The abundance of *Ruminococcaceae* in rats. (**G**) The abundance of *Ruminococcus* in rats. (**H**) The abundance of *Streptococcus* in rats. (**I**) The abundance of *Turicibacter* in rats. (**J**) Abundance clustering heatmap of these nine genera. The data are presented as the mean ± SD, n = 6. ** *p* < 0.01 vs. WKY; **^##^** *p* < 0.01 vs. untreated SHR.

**Figure 9 molecules-28-00612-f009:**
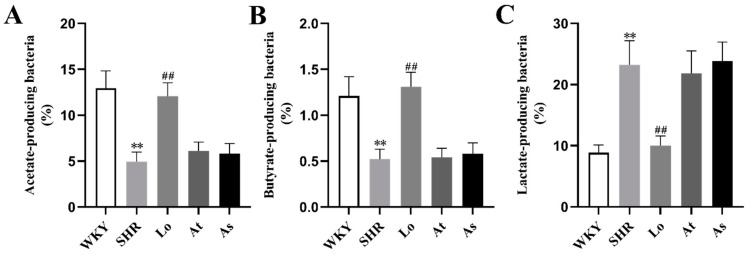
The abundances of acetate-, butyrate-, and lactate-producing bacteria in the gut microbiota in rats. (**A**) The abundance of acetate-producing bacteria in the gut microbiota in rats. (**B**) The abundance of butyrate-producing bacteria in the gut microbiota in rats. (**C**) The abundance of lactate-producing bacteria in the gut microbiota in rats. The data are presented as the mean ± SD, n = 6. ** *p* < 0.01 vs. WKY; **^##^** *p* < 0.01 vs. untreated SHR.

**Figure 10 molecules-28-00612-f010:**
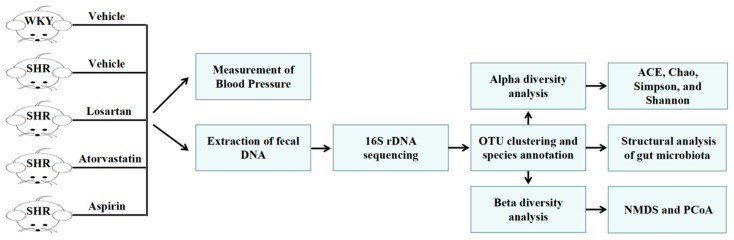
Experimental flow chart of this study.

**Table 1 molecules-28-00612-t001:** The abundance of the five most abundant phyla in the gut microbiota in rats.

Phylum	WKY	SHR	Lo	At	As
Actinobacteria (%)	1.88 ± 0.43	0.29 ± 0.04 **	1.50 ± 0.23 **^##^**	0.43 ± 0.10	0.34 ± 0.06
Bacteroidetes (%)	42.22 ± 5.37	24.79 ± 3.99 **	46.22 ± 4.72 **^##^**	25.95 ± 3.00	22.53 ± 3.20
Firmicutes (%)	53.10 ± 5.18	70.14 ± 3.83 **	48.75 ± 5.13 **^##^**	68.80 ± 3.06	74.43 ± 3.33
Proteobacteria (%)	1.80 ± 0.31	2.00 ± 1.19	1.92 ± 0.69	2.39 ± 1.08	2.16 ± 0.27
Tenericutes (%)	0.21 ± 0.06	0.25 ± 0.14	0.22 ± 0.09	0.20 ± 0.09	0.18 ± 0.07

The data are presented as the mean ± SD, n = 6. ** *p* < 0.01 vs. WKY; **^##^** *p* < 0.01 vs. untreated SHR.

## Data Availability

Not applicable.

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
