# Peer review of "Effects of Losartan, Atorvastatin, and Aspirin on Blood Pressure and Gut Microbiota in Spontaneously Hypertensive Rats"

_molecules, 2023, doi:10.3390/molecules28020612_

Round 1

Reviewer 1 Report

The study aimed to observe the characteristics of gut microbiota in hypertension, and to further explore whether drug molecules can play a therapeutic role in hypertension by interfering with gut microbiota. But there are some problems need to be clarified.

1.Although losartan increased the abundance of Alistipes, Bacteroides and Butyricimonas in SHR, reduced the abundance of Ruminococcaceae, Streptococcus and Turicibacter, reduced the Firmicutes/Bacteroidetes ratio, and rebalanced the gut microbiota. Its more believable to study their efficacy in free-bacteria mice or single bacterium transplanted mice to demonstrate that they lower blood pressure by changing the characteristics of gut microbiota in hypertension and rebalance the gut microbiota.

2.Figure 10 list the structural formula of the three drugs, whats the relationship of the structural formula and the gut microbiota regulation?

3. There should be uniformity in indicating the various groups in both text as well as figures. E.g. SHR-losartan, SHR-Atorvastatin, and SHR-Aspirin groups has been used in text while Lo, At and As has been used in figure 1 and figure 3. Similar is the case for other groups. Please rectify.

4. Full form of many abbreviations is not given for the first time. Some of the examples are Lo, At, As, OTU, ACE, NMDS and PCoA etc. Italic is neglected, such as P.

5. The unit of number has been neglected, some of the examples are Table 1, Figure1, Figure 2, Figure8 and 9.

6. Improve figure legend, bring the abbreviation names

7. Explain Proteobacteria and Tenericutes in Table 1 and Figure 6.

8.Please describe the results of Figure 8D-8I.

9. Explain better what we should see in 9a, 9b, 9c in the different parts. The results are not consistent with the legend of figures. Are they abundances of acetate-, butyrate-, and lactate-producing bacteria or acetate,butyrate and lactate?.

10.In "Materials and Methods", significant details are missing, nor are any references given for these details.

Reviewer 2 Report

This is a sound and interesting paper, is well written and English is appropriate. The Reference section seems updated.

Nevertheless i have one major concern regarding the conclusions and (possibly) a second major concern regarding the methods:

(1) on the basis of the data here presented the conclusions of the authors about a possible antihypertensive effects of losartan mediated by microbiota changes is completelly unjustified. Authors have simply observed differences between WKY and SHR rats in terms of gut microbiota and they have observed that under losartan treatment these differences change, reaching similar values in SHR (losartan-treated) vs. WKY. So the statements advanced at page 12 (rows 294-295) and page 14 (rows 365-366 and 373) seem to me an overinterpretation of the data. Simply pretending a cause-effect ratio unjustified by the methodological approach;

(2) methodologically speaking the authors have administered losartan obtained from pharma industry, I suppose dispersed in vehicle. If this (as logical) is the case, the right control should be drug against vehicle alone and not water alone, since even the vehicle can induce possible microbiota changes.

I believe that the discussion should be changed profoundly and even the experimental procedure should be changed before an evaluation for publication.

Reviewer 3 Report

In this study, taking spontaneously hypertensive rats (SHR) as the observation object, the characteristics of the intestinal flora of hypertensive patients and the effects of three cardiovascular drugs, losartan, atorvastatin and aspirin, on the intestinal flora of SHR were investigated. From the perspective of intestinal flora, it provides a basis for supporting the treatment of hypertension. Suggested for publication, the article still has some problems.

1. The heat map below Figure 8 corresponding to the content of the article is not clear, please mark it in the article.

2. Can the authors clarify in the Discussion section what implications the findings of this study have for guiding clinical use and future research directions.

3. It is recommended to add an experimental flow chart to the article so that readers can better understand the overall idea of the article.

Round 2

Reviewer 1 Report

Acceptance after minor spell check required.

Reviewer 2 Report

I am afraid to say that the response of the authors do not resolve the concerns I find in the manuscript, so I believe the paper is unacceptable for publication on this journal
